

# Occurrence of termites (Isoptera) on living and standing dead trees in a tropical dry forest in Mexico

Nancy Calderón-Cortés[1], Luis H. Escalera-Vázquez[2] and Ken Oyama[1]

[1] Escuela Nacional de Estudios Superiores Unidad Morelia, Universidad Nacional Autónoma de México, Morelia, Michoacán, México

[2] CONACyT, Instituto de Investigaciones sobre los Recursos Naturales, Universidad Michoacana de San Nicolás de Hidalgo, Morelia, Michoacán, México

## ABSTRACT

Termites play a key role as ecosystem engineers in numerous ecological processes though their role in the dynamics of wood degradation in tropical dry forests, particularly at the level of the crown canopy, has been little studied. In this study, we analysed the occurrence of termites in the forest canopy by evaluating the density and proportion of living and standing dead trees associated with termites in deciduous and riparian habitats of the tropical dry forest in Chamela, Mexico. The results indicated that 60–98% of standing dead trees and 23–59% of living trees in Chamela were associated with termites. In particular, we found that the density of standing dead trees was higher in deciduous forests ($0.057$–$0.066$ trees/m$^2$) than in riparian forests ($0.022$ and $0.027$ trees/m$^2$), even though the proportion of trees was not significantly different among habitats. Additionally, we found a higher density of trees associated with termites in trees of smaller size classes ($0.01$–$0.09$ trees/m$^2$) than in larger class sizes ($0$–$0.02$ trees/m$^2$). Interestingly, 72% of variation in the density of trees associated with termites is explained by the density of standing dead trees. Overall, these results indicate that standing dead tree availability might be the main factor regulating termite populations in Chamela forest and suggest that termites could play a key role in the decomposition of above-ground dead wood, mediating the incorporation of suspended and standing dead wood into the soil.

## INTRODUCTION

The decomposition of organic matter stands out as a central component of ecosystem functioning, playing important roles related to nutrient cycling and energy flow, and influencing the diversity in ecosystems (*Murphy & Lugo, 1986*). Termites (Blattaria: Isoptera) represent one of the most important groups of organisms that participate in organic matter decomposition in tropical ecosystems since they constitute between 40 to 95% of the total biomass of soil macrofauna (*Dangerfield, McCarthy & Ellery, 1998*; *Donovan et al., 2007*). The role of termites in ecosystem processes in tropical forests has

Corresponding author
Nancy Calderón-Cortés,
ncalderon@enesmorelia.unam.mx

been widely documented, demonstrating that termites regulate the incorporation and decomposition of organic matter to the soil, contribute to carbon and nitrogen mineralization, alter vegetation composition, and improve soil structure by increasing porosity, aeration, and water retention (*Aanen et al., 2002*; *Dawes-Gromadzki, 2003*; *Ohkuma, 2003*; *Jeyasingh & Fuller, 2004*; *Donovan et al., 2007*; *Erpenbach et al., 2013*; *Dahlsjö et al., 2014*; *Houston, Wormington & Black, 2015*; *Maynard et al., 2015*). Therefore, in these ecosystems, termites are recognized as keystone ecosystem engineers that directly or indirectly modify the availability of nutrients for other organisms through the decomposition of plant material (*Dangerfield, McCarthy & Ellery, 1998*; *Bignell, 2006*; *Moe, Mobæk & Narmo, 2009*; *Jouquet et al., 2011*; *Romero et al., 2014*), particularly when termites incorporate dead woody material suspended in the forest canopy (i.e., standing dead trees and fallen trunks and branches) that is otherwise inaccessible to soil micro- and macro-fauna (*Maynard et al., 2015*).

Coarse woody debris (CWD) is an important dead-wood component of tropical forests, and therefore of the decomposition process, that can represent as much as 42% of the above-ground biomass (*Clark et al., 2004*; *Rice et al., 2004*; *Pfeifer et al., 2015*). The decomposition of woody material is usually considered to be primarily a forest-floor process because the contact of plant debris with organisms in the soil, as well as with the suitable micro-environmental conditions of the soil interface, increases the decomposition rates (*Harmon et al., 1986*). In contrast, it is assumed that higher irradiance and windy conditions in the upper canopy create a drier environment that limits the type and abundance of decomposers, thereby reducing the rate of decomposition of woody material (*Harmon et al., 1986*; *Fonte & Schowalter, 2004*). However, in some subtropical dry forests, it has been found that approximately three-quarters of the available pieces of dead wood in the forest canopy (i.e., standing dead trees and suspended dead wood in the crown of trees) present signs of termite activity (*Jones et al., 1995*). Indeed, the role of termites in decomposing standing dead wood has been recognized for nearly 100 years in temperate forests (*Maynard et al., 2015*, and references therein), but its role in the decomposition process of dead woody material at the canopy level (i.e., dead wood above-ground) has been poorly studied in tropical forests. In particular, the role of termites in wood decomposition has not been investigated in tropical forests of Mexico (*Méndez-Montiel & Martínez-Equihua, 2001*; *Maass et al., 2002*; *Rodríguez-Palafox & Corona, 2002*).

In this study, we recorded the presence of termites on living and standing dead trees in a tropical dry forest in Chamela, Jalisco, to illuminate the role that termites could play in the decomposition of above-ground dead woody material of a tropical dry forest in Mexico. The diversity of termites in the Chamela forest is represented by 30 species belonging to three families: Kalotermitidae (10 spp.), Rhinotermitidae (two spp), and Termitidae (18 spp), including species feeding on dry wood, decaying wood and humus (i.e., soil feeders) (*Nickle & Collins, 1988*; *Rodríguez-Palafox & Corona, 2002*). Based on independent annual estimations, it is known that dead woody material in the tropical dry forest in Chamela comprises approximately 32% of total biomass above-ground (*Duran et al., 2002*; *Segura et al., 2003*). From this, 20–53% represent dead wood in the forest floor, and 46–80% represent standing dead trees and suspended branches (*Duran et al., 2002*;

 

*Maass et al., 2002*; *Segura et al., 2003*), indicating that decomposition of suspended and standing dead wood is critical for this forest. The presence of all major termite groups in Chamela, including wood dwelling and carton-nest building species (*Nickle & Collins, 1988*), and the occurrence of termite galleries in most of the standing dead trees, suggest that termites play an important role in decomposing wood in the forest canopy (*Maass et al., 2002*). To our knowledge, the present study represents the first approach to analyse the occurrence of termites on living and standing dead trees to illuminate the potential role that termites play in the decomposition of above-ground dead woody material in a tropical dry forest in Mexico. In particular, we addressed the following questions: (i) What is the density of trees associated with termites?; (ii) Is the density of living and standing dead trees associated with termites different between deciduous and riparian habitats?; (iii) What is the proportion of living and standing dead trees associated with termites?; (iv) Are these proportions different between habitats?; (v) Does tree size determine the association with termites?; and (vi) Does the density of standing dead trees explain the presence of termites in trees?

## MATERIALS AND METHODS

### Study site

The study was conducted during the rainy season (October) in 2004 and 2009 at the Chamela Biological Station, UNAM (19°30′N, 105°03′W), located in the state of Jalisco, Mexico, 2 km east of the Pacific coast. The climate is warm with a mean annual temperature of 24.6 °C and an annual rainfall of 748 mm (*Lott, Bullock & Solis-Magallanes, 1987*). This ecosystem presents a marked seasonality, with a dry season from November to May (when the forest canopy is lost) and a rainy season from June to October (*García-Oliva, Camou & Maass, 2002*). The dry forest, located on upland soils, is dominated by deciduous trees 4–15 m in height, with a well-developed understory of shrubs and some patches of tropical riparian forest associated with intermittent streams (*Rzedowski, 1978*; *Lott, Bullock & Solis-Magallanes, 1987*). The landscape consists of low hills (50–160 m elevation) with steep convex slopes with alluvial and sandy soils of variable deep (*García-Oliva, Camou & Maass, 2002*; *Jaramillo et al., 2003*).

### Methods

We sampled five plots of 200 m² (4 × 50 m) in each habitat (i.e., deciduous and riparian forest) for each year. The plots were randomly selected next to the paths in the biological station but oriented perpendicular to the path to reduce the border effect. Within each plot, we registered living and standing dead trees, with and without termite activity signs (i.e., the presence of galleries). The diameter at breast height (DBH = 1.3 m) of each tree was measured; for multi-stemmed trees, DBH was calculated as the sum of each stem measure. The collected termite samples were conserved in ethanol 70% for further identification to the lowest taxonomic level based on *Nickle & Collins (1988)*.

### Data analysis

We estimated the density of living and standing dead trees with and without termites in each plot as the total number of trees divided by the plot area (200 m²). To evaluate the
effect of habitat on the density trees with and without termites, we analysed the density of living and standing dead trees using the GENMOD procedure with a Poisson distribution and logarithmic link function. The model used tree density as the dependent variable and habitat (deciduous and riparian), year and the interaction between habitat and year as independent variables.

To determine the proportion of trees associated with termites, we registered the presence/absence of termites in living and standing dead trees in the two habitats. Data were analysed using the GENMOD procedure, with a binomial distribution, logistic link function and descending option (i.e., modelling the probability of termite presence). The presence of termites was used as the dependent variable; meanwhile, habitat, year and the interaction between habitat and year were fixed as independent variables. DBH was used as a covariate.

In addition, to evaluate the effect of tree size on the density of living and standing dead trees with and without termites, we classified all trees inside plots into different size classes: I saplings (DBH $\leq$ 5 cm); II juveniles (DBH > 5 $\leq$ 15 cm); III medium adults (DBH > 15 $\leq$ 25 cm); and IV large adults (DBH > 25 cm). The density of living and standing dead trees with and without termites was analysed using the GENMOD procedure with a Poisson distribution and logarithmic link function for each year; density was used as the dependent variable, while size class (saplings, juveniles, medium adults and large adults) and the interaction between size class and habitat were used as independent variables. Finally, we performed a linear regression analysis (GLM procedure) to evaluate whether the density of standing dead trees is related to the density of trees associated with termites. All statistical analyses were performed using SAS software (*SAS Institute , 2017*).

## RESULTS

### Termites and habitat parameters

The termite species collected on trees in the Chamela forest were *Nasutitermes nigriceps, N. mexicanus* and *Amitermes* spp (Termitidae). The Nasutitermitinae species were frequently collected from galleries in both living and standing dead trees and from cartoon nests, while *Amitermes* spp. were collected from the interior of standing dead trees.

Our results showed that habitat ($F_{1,16} = 2.21$, $P = 0.1569$) and habitat*year interaction ($F_{1,16} = 0.11$, $P = 0.7423$) did not affect the density of living trees, although the density of living trees was significantly higher in 2009 than in 2004 ($F_{1,16} = 16.97$, $P = 0.0008$; Table 1). The opposite pattern was found for standing dead trees: habitat significantly affected tree density ($F_{1,16} = 9.14$, $P = 0.0081$), but neither year ($F_{1,16} = 3.98$, $P = 0.0633$) nor the habitat*year interaction ($F_{1,16} = 0.26$, $P = 0.6177$) had a significant effect on tree density. Deciduous forest showed a higher density of standing dead trees than riparian forest (Table 1).

We also found that tree size class had a significant effect on the density of both living ($F_{3,32} = 11.00, P < 0.0001$) and standing dead trees ($F_{3,32} = 5.60, P = 0.003$). High densities of living trees were found in smaller size classes (I and II), but the size class with the highest tree density differed in each year: size class I in 2004, and size class II in 2009 (Table 1). In

**Table 1   Habitat parameters.**

| | 2004 | | 2009 | |
|---|---|---|---|---|
| | **Deciduous forest** | **Riparian forest** | **Deciduous forest** | **Riparian forest** |
| **Living trees** | | | | |
| Total density | $0.164 \pm 0.058^{ab}$ | $0.090 \pm 0.044^{a}$ | $0.513 \pm 0.100^{cd}$ | $0.352 \pm 0.085^{bc}$ |
| Size class I density | $0.046 \pm 0.017^{ab}$ | $0.025 \pm 0.012^{bc}$ | $0.268 \pm 0.041^{A}$ | $0.202 \pm 0.036^{A}$ |
| Size class II density | $0.070 \pm 0.021^{a}$ | $0.042 \pm 0.016^{ab}$ | $0.184 \pm 0.034^{A}$ | $0.106 \pm 0.026^{AB}$ |
| Size class III density | $0.026 \pm 0.013^{bd}$ | $0.009 \pm 0.007^{c}$ | $0.036 \pm 0.015^{B}$ | $0.029 \pm 0.031^{BC}$ |
| Size class IV density | $0.018 \pm 0.010^{cd}$ | $0.014 \pm 0.009^{cd}$ | $0.025 \pm 0.012^{BC}$ | $0.015 \pm 0.009^{C}$ |
| **Standing dead trees** | | | | |
| Total density | $0.060 \pm 0.013^{ab}$ | $0.025 \pm 0.008^{c}$ | $0.087 \pm 0.016^{a}$ | $0.047 \pm 0.012^{b}$ |
| Size class I density | $0.019 \pm 0.006^{ac}$ | $0.008 \pm 0.004^{bc}$ | $0.031 \pm 0.007^{A}$ | $0.013 \pm 0.005^{AB}$ |
| Size class II density | $0.026 \pm 0.068^{a}$ | $0.011 \pm 0.004^{bc}$ | $0.048 \pm 0.009^{A}$ | $0.028 \pm 0.007^{A}$ |
| Size class III density | $0.010 \pm 0.004^{bc}$ | $0.003 \pm 0.002^{b}$ | $0.003 \pm 0.002^{B}$ | $0.005 \pm 0.003^{B}$ |
| Size class IV density | $0.005 \pm 0.003^{b}$ | $0.002 \pm 0.001^{b}$ | $0.005 \pm 0.003^{B}$ | $0.001 \pm 0.001^{B}$ |

\*Density values are expressed in tree/m². Different letters indicate significant differences ($P < 0.05$). Size classes: I, saplings (DBH $\leq$ 5 cm); II, juveniles (DBH $> 5 \leq 15$ cm); III, medium adults (DBH $> 15 \leq 25$ cm); IV, large adults (DBH $> 25$ cm).

the case of standing dead trees, size class II show the highest tree density, followed by size class I (Table 1). This pattern was consistent across years. The size class *habitat interaction was not significant in any year for both living and standing dead trees (2004: $F_{3,32} = 0.46$, $P = 0.7126$ and $F_{3,32} = 0.06$, $P = 0.9824$; 2009: $F_{3,32} = 0.11$, $P = 0.9518$ and $F_{3,32} = 0.65$, $P = 0.5909$, respectively).

## Interaction of termites with trees

There was a significant difference in the density of living trees associated with termites between habitats ($\chi^2 = 8.15$, $P = 0.0043$) and years ($\chi^2 = 8.15$, $P = 0.0043$), but the habitat*year interaction did not show a significant difference ($\chi^2 = 0.69$, $P = 0.4071$). The density of living trees associated with termites was higher in deciduous (0.091 and 0.19 trees/m² for 2004 and 2009, respectively) than in riparian forest (0.061 and 0.091 trees/m² for 2004 and 2009, respectively; Fig. 1A), although a significant difference was only found in 2009 (Fig. 1A). Similarly, the density of standing dead trees associated with termites showed a significant difference between habitats ($\chi^2 = 15.29$, $P < 0.0001$), with a higher density in deciduous (0.057 and 0.066 trees/m² for 2004 and 2009, respectively) than in riparian forest (0.022 and 0.027 trees/m² for 2004 and 2009, respectively; Fig. 1B), although there was no significant effect of year ($\chi^2 = 0.5$, $P = 0.4788$) or the habitat*year interaction ($\chi^2 = 0.01$, $P = 0.9068$) (Fig. 1B).

In total we registered the presence/absence of termites in 1,334 trees. Our results indicated that habitat did not have a significant effect on the proportion of living trees associated with termites ($\chi^2 = 0.02$, $P = 0.8986$; Fig. 2A). However, year ($\chi^2 = 55.06$, $P < 0.0001$), the habitat*year interaction ($\chi^2 = 10.02$, $P = 0.0016$), and DBH ($\chi^2 = 20.05$, $P < 0.0001$) showed a significant effect on the proportion of living trees associated with termites, with a greater percentage of living trees associated with termites in 2004 than in 2009 in both habitats (Fig. 2A). On the other hand, the proportion of standing dead

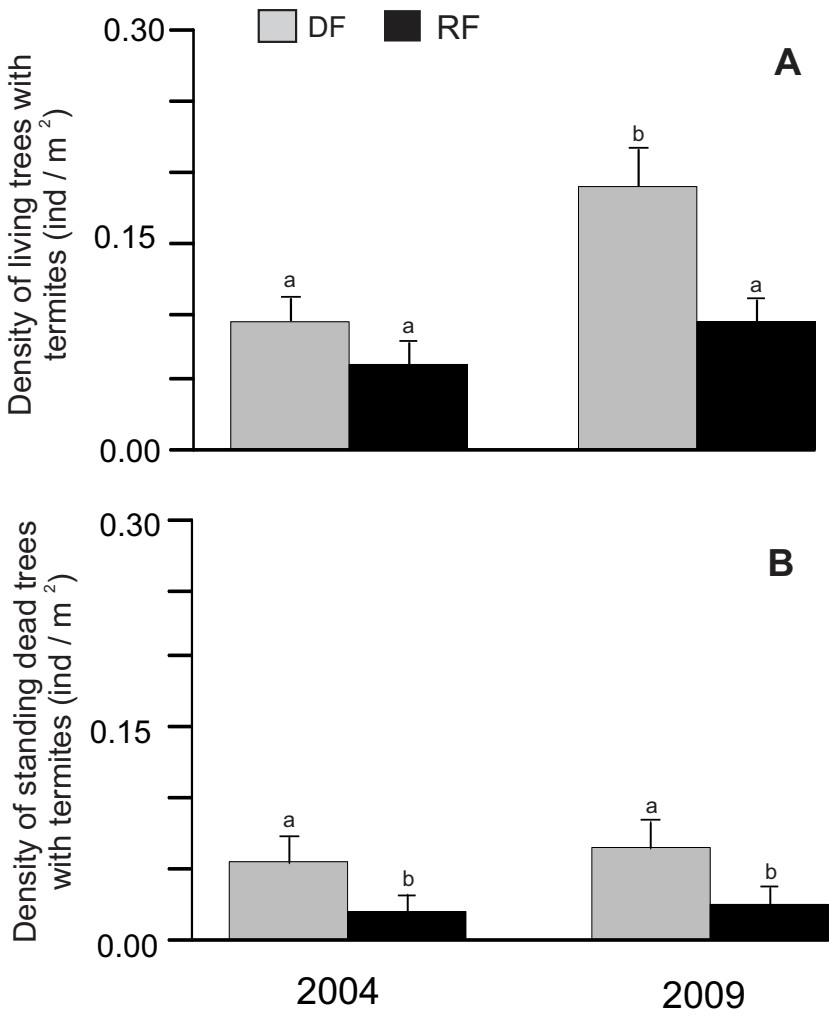

**Figure 1  Density of trees associated with termites in Chamela.** (A) Comparison of the density of living trees with termites in two habitats. (B) Comparison of the density of standing dead trees with termites in two habitats. LSMeans (±SE) are shown in bars: grey bars for deciduous forest (DF); black bars for riparian forest (RP). Density values are reported as individuals per m². Different letters indicate significant differences ($P < 0.005$) between the density across habitats and years.

trees associated with termites was significantly affected by year ($\chi^2 = 16.05$, $P < 0.0001$) and DBH ($\chi^2 = 4.43$, $P = 0.0353$), but there was no significant effect by habitat ($\chi^2 = 3.8$, $P = 0.0512$) or by the habitat*year interaction ($\chi^2 = 0.03$, $P = 0.8729$) (Fig. 2B). Similar to living trees, the proportion of standing dead trees associated with termites was greater in 2004 than in 2009 (Fig. 2B).

Size class showed a significant effect on the density of both living and standing dead trees associated with termites in both years (2004: $F_{3,32} = 10.56$, $P < 0.0001$ and $F_{3,32} = 5.01$, $P = 0.0058$; 2009: $F_{3,32} = 8.58$, $P = 0.0003$ and $F_{3,32} = 16.54$, $P < 0.0001$ for living and standing dead trees, respectively). However, the size class*habitat interaction did not affect either the density of living trees or the density of standing dead trees in any year. In general,

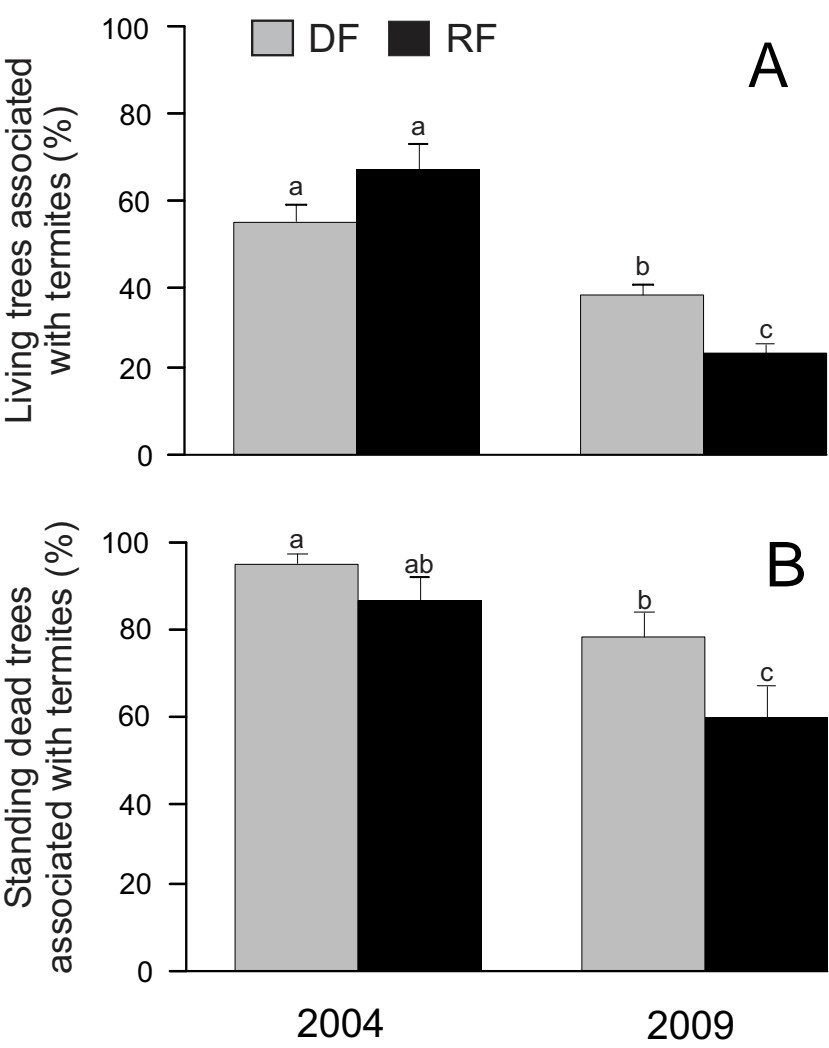

**Figure 2 Proportion of trees associated with termites in Chamela.** (A) Comparison of the proportion of living trees with termites in two habitats. (B) Comparison of the proportion of standing dead trees with termites in two habitats. LSMeans (±SE) are shown in bars: grey bars for deciduous forest (DF); black bars for riparian forest (RP). Values are the percentage of trees with termites. Different letters indicate significant differences ($P < 0.005$) between the density across habitats and years.

the density of living and standing dead trees was significantly higher in size class II, followed by size class I, size class III and size class IV, in both habitats and years (Figs. 3A and 3B).

Finally, our results indicated that the density of standing dead trees was positively related to the density of trees associated with termites ($R^2 = 0.72$, $F_{1,20} = 47.65$, $P < 0.0001$; Fig. 4).

## DISCUSSION

Termites are among the invertebrates with the largest biomass and abundance in tropical forests (*Bignell & Eggleton, 2000*; *Vasconcellos, 2010*). However, due to the great diversity of microhabitats they occupy (i.e., soil, decomposing wood, leaf litter, arboreal or soil mounds, inside the living trees and/or fallen branches that remain in the forest canopy), it

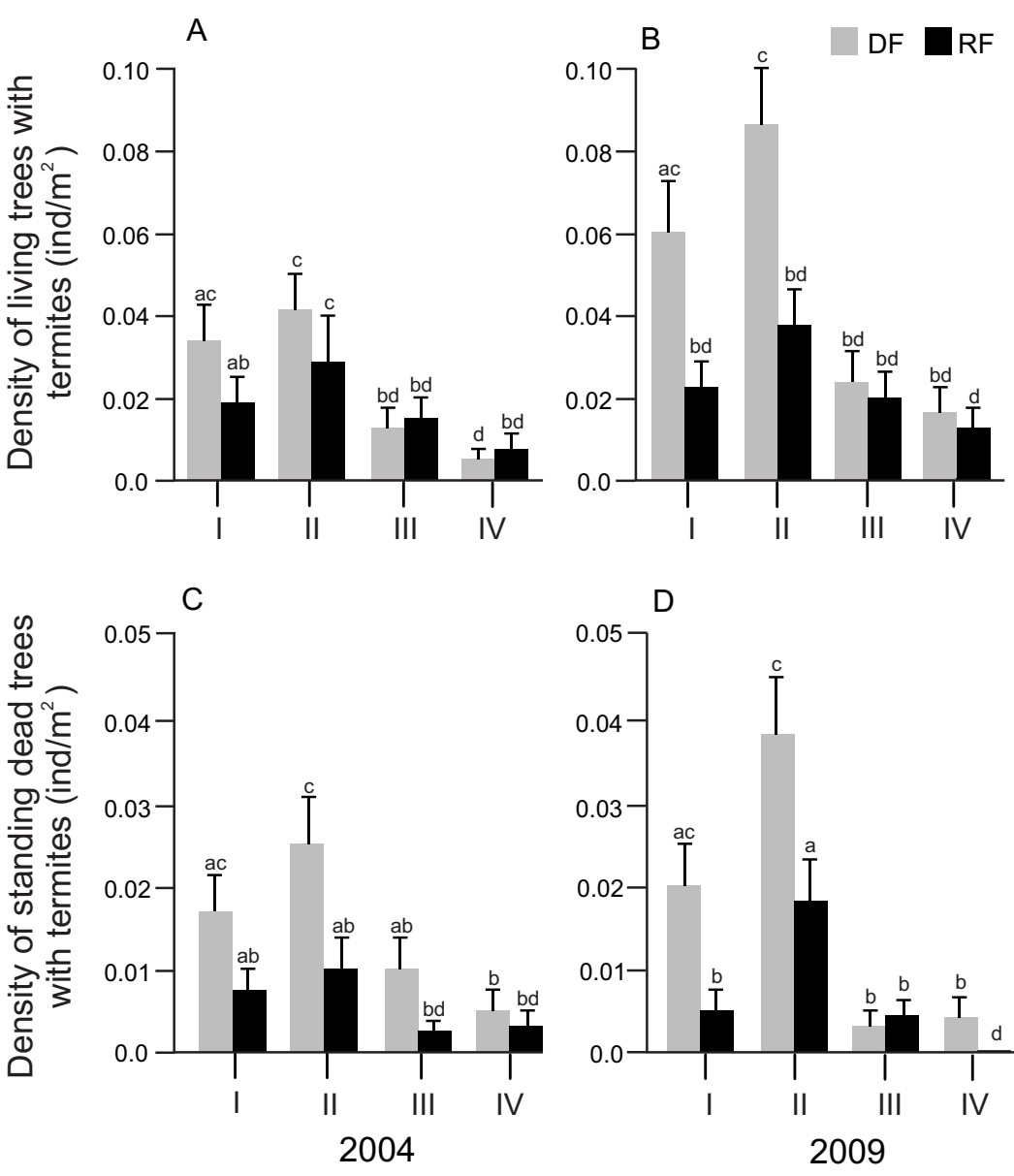

**Figure 3** **Distribution of the density of trees associated with termites in different diameter size classes in Chamela.** (A) Density of living trees with termites in 2004. (B) Density of living trees with termites in 2009. (C) Density of standing dead trees with termites in 2004. (D) Density of standing dead trees in 2009. LSMeans (±SE) are shown in bars: grey bars for deciduous forest (DF); black bars for riparian forest (RP). Diameter size classes are indicated by Roman numerals: I, saplings (DBH ≤ 5 cm); II, juveniles (DBH > 5 ≤ 15 cm); III, medium adults (DBH > 15 ≤ 25 cm); IV, large adults (DBH > 25 cm). Statistical analyses were performed for each year (2004 and 2009). Different letters indicate significant differences ($P < 0.005$) across size classes and habitats.

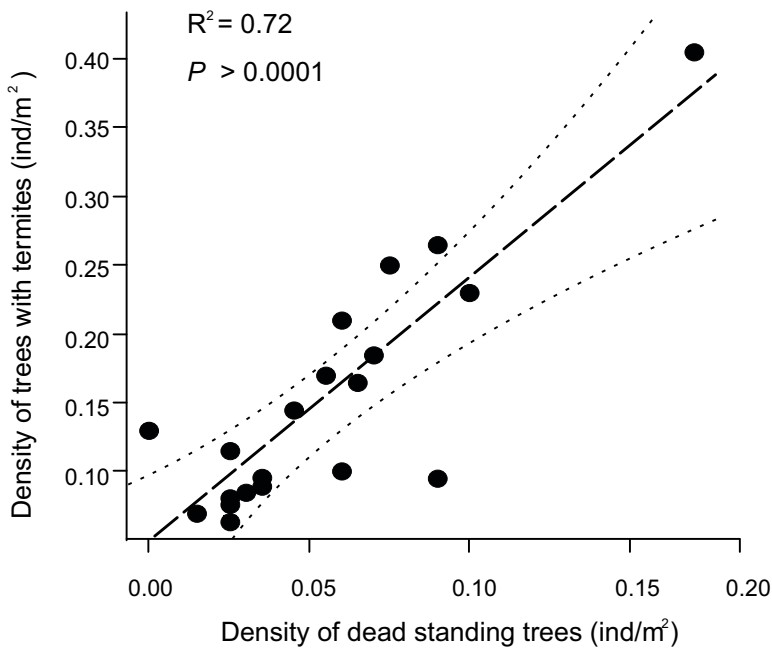

**Figure 4** **Relationship between the density of trees associated with termites and the density of standing dead trees in Chamela.** Density was estimated as individuals per m$^2$. Grey dashed lines indicate the 95% confidence interval. The estimated linear regression equation is $y = 1.9126x + 0.0465$ ($n = 20$).

is difficult to evaluate the density of these insects in the forest (*Eggleton et al., 1995*). Termite density has been estimated based on the number of encounters with termites in transects of 200 m$^2$ (*Davies, 1997*; *Eggleton et al., 1995*; *Eggleton et al., 1996*; *Crist, 1998*; *Inoue et al., 2006*), based on nest volume (*Jeyasingh & Fuller, 2004*) and in controlled experiments using baits (*Dawes-Gromadzki, 2003*; *Davies et al., 2015*), although these estimations might not reflect the potential of colonization of dead wood available in the forest canopy. In this study, we evaluated the occurrence of termites on living and standing dead trees in a tropical dry forest based on the density and proportion of trees in which termites are present, and in general, our results indicate that the availability of standing dead trees is associated with the density of trees with termites.

Specifically, our results showed that the proportion of living and standing dead trees was not different among habitats, indicating that habitat characteristics might not affect the termite presence on trees. Nevertheless, we observed a general pattern of higher density of trees with termite activity in deciduous than in riparian forest for both living and standing dead trees. This pattern can be explained by the difference in resource availability (i.e., necromass) for termites, which in turn can be related to the density and volume of dead wood present in each habitat. Periodic inundations of riparian habitats can also control termite populations since flooding events reduce termite populations, particularly subterranean termites (*Ulyshen, 2014*). However, it is unknown whether flooding events can negatively affect arboreal termites such as Termitidae species.

Necromass includes either suspended and soil wood materials, as well as standing dead material (*Harmon et al., 1986*). However, standing dead trees or snags constitute an important component of necromass (46–80%) in Chamela forest (*Duran et al., 2002*; *Maass et al., 2002*). Interestingly, our results indicate that 60–98% of standing dead trees in Chamela were associated with termites, and similar to a previous study evaluating the density and proportion of standing dead trees in Chamela (*Segura et al., 2003*), our results also indicated that the density of standing dead trees was higher in deciduous (600–870 trees/ha) than in riparian forest (250–470 trees/ha). In addition, the density of both living and standing dead trees associated with termites was also higher in trees of smaller class sizes (<15 cm DBH), for which we also found a higher density of standing dead trees. High mortality rates in small trees (<10 cm of DBH) have been reported for other tropical ecosystems (*Bellingham & Tanner, 2000*; *Lorimer, Dahir & Nordheim, 2001*; *Clark et al., 2004*). Overall, these results suggest that standing dead tree availability might be the main factor regulating termite occurrence in Chamela forest. This hypothesis was confirmed by the positive relationship found between the density of standing dead trees and the density of trees associated with termites ($r^2 = 0.72$). There is available evidence indicating positive correlations between the volume of dead wood and the volume of termite nests (i.e., as a surrogate of termite density; *Jones et al., 1995*; *Jeyasingh & Fuller, 2004*), but interestingly, a positive correlation between standing dead wood and termite density at a broad regional scale (i.e., the USA) has been recently reported (*Maynard et al., 2015*). A positive relationship between dead wood and insect density has also been reported for saproxylic borer beetles (*Grove, 2002*; *Lachat et al., 2012*).

Tree mortality and dead wood production (i.e., necromass) is episodic and varies greatly over temporal and spatial scales (*Martius, 1997*; *Palace et al., 2012*). Several factors and mechanisms have been reported to explain this variation (i.e., tree competition for nutrients and light, topography, root system characteristics of trees, among others), but disturbance appears to play a prominent role (*Martínez-Ramos et al., 1988*; *Gale, 2000*; *Palace et al., 2012*). In this sense, habitats subjected to frequent disturbance are expected to have high necromass production (*Palace et al., 2012*). In Chamela forest, such disturbance events are reported to be associated with drought followed by windthrow related to frequent tropical summer storms (*García-Oliva, Maass & Galicia, 1995*), but it is recognized that drought events play a key role in tree mortality in this forest, particularly at higher elevation sites, such as deciduous forests, where the driest conditions prevailed (*García-Oliva, Maass & Galicia, 1995*; *Duran et al., 2002*; *Maass et al., 2002*), explaining the high proportion of standing dead trees we found in deciduous forest. However, our results also indicated that termites are present in a considerable proportion of living trees (23–59%), suggesting that termites present in these trees are using dead branches suspended in the forest canopy. The decomposition of both standing and suspended dead wood is crucial in the dynamics of nutrient and energy flux in the ecosystem (*Harmon et al., 1986*). As we mentioned before, in Chamela forest, the biomass of standing dead trees and suspended branches (46–80%) exceeds the biomass of dead wood in the forest floor (20–53%), and in some sites, it exceeds the biomass of litter (*Maass et al., 2002*; *Jaramillo et al., 2003*), indicating that the decomposition of suspended and standing dead wood is critical for this forest

(*Maass et al., 2002*). Standing and suspended dead wood are also an important component of total necromass in other tropical ecosystems, reaching 66% in undisturbed forests and 98% in sites with high disturbance levels (review in *Palace et al., 2012*).

Undoubtedly, abiotic factors such as strong winds and hurricanes can enhance the incorporation of suspended dead wood to the forest floor (*Harmon et al., 1995*; *Maass et al., 2002*). However, the presence of termites, fungi and saproxylic beetles have been suggested as the main factors related to the decomposition process of dead wood in the forest canopy (*Swift et al., 1976*; *Fonte & Schowalter, 2004*; *Maynard et al., 2015*). The abundance of termites and decay rates of dead wood in the canopy have not been investigated in Chamela forest (*Maass et al., 2002*), but the presence of termites on a great proportion of living and standing dead trees found in this study suggests that termites could be an important decay agent at the canopy level in Chamela forest. Similar results were reported in subtropical dry forests in Mona Island, Puerto Rico, where nearly three-quarters of the available pieces of dead wood showed signs of termite attack (*Jones et al., 1995*). The decay process in the forest is important in determining carbon and nutrient inputs into the ecosystem, but early stages of decay occurring at the canopy level are particularly important since they determine the rate of wood debris incorporation into the soil, with long-term effects on nutrient ecosystem dynamics (*Fonte & Schowalter, 2004*; *Maynard et al., 2015*). Termites can be responsible for a considerable proportion of carbon mineralization (i.e., 20%) and nitrogen input to the soil (i.e., as high as 0.5 kg N/ha per year: reviewed in *Maynard et al., 2015*). In Chamela, 39% and 49% of C and N forest stocks are sequestered in total above-ground necromass (*Jaramillo et al., 2003*), and given that termites are frequently present in standing dead trees of this forest, it is possible that they contribute significantly to C and N cycling.

## CONCLUSIONS

The high proportion of trees on which termites are present in Chamela tropical dry forest, the patterns of density of trees with termites in different habitats and size classes of trees that vary in necromass availability, and the positive relationship between the density of trees associated with termites and the density of standing dead trees suggest that termites could be important agents in wood decomposition at the canopy level in this forest. However, given that this study did not estimate the abundance and/or degradation decay rates mediated by termites and that this insect group has not been studied in tropical forests in Mexico, future studies that quantify wood decomposition by termites at the canopy and floor levels and that estimate the abundance and diversity patterns of termites in different habitats (specially in those habitats experienced changes in land use) are needed to understand the role that termites play in tropical forests in Mexico, particularly in a scenario of global change that threatens biodiversity and ecosystem processes.

## ACKNOWLEDGEMENTS

The authors thank the Postgraduate Program of Biological Sciences of the UNAM, M Quesada and Y Herrerías-Diego for valuable comments and suggestions during field work,

and FO Saavedra-Cazáres and TG Marques-Silva for field work support. Finally, we thank the Chamela Biological Station, UNAM for facilities and logistical support.

### Funding

This work was supported by Posgrado en Ciencias Biologicas, UNAM (as part of an Ecology field course), and by grants from CONACyT (CB-2015-253420 to Nancy Calderón-Cortés) and Universidad Nacional Autónoma de México (DGAPA-UNAM, PAPIIT # IA200918 to Nancy Calderón-Cortés). The funders had no role in study design, data collection and analysis, decision to publish, or preparation of the manuscript.

### Grant Disclosures

The following grant information was disclosed by the authors:
Posgrado en Ciencias Biologicas, UNAM (as part of an Ecology field course).
CONACyT: CB-2015-253420.
Universidad Nacional Autónoma de México: DGAPA-UNAM, PAPIIT # IA200918.

### Competing Interests

The authors declare there are no competing interests.

### Author Contributions

- Nancy Calderón-Cortés conceived and designed the experiments, performed the experiments, analyzed the data, contributed reagents/materials/analysis tools, prepared figures and/or tables, authored or reviewed drafts of the paper, approved the final draft.
- Luis H. Escalera-Vázquez performed the experiments, analyzed the data, contributed reagents/materials/analysis tools, prepared figures and/or tables, authored or reviewed drafts of the paper, approved the final draft.
- Ken Oyama analyzed the data, contributed reagents/materials/analysis tools, authored or reviewed drafts of the paper, approved the final draft.

### Data Availability

 The raw data are provided as Supplemental File.

### Supplemental Information

Supplemental information for this article can be found online at http://dx.doi.org/10.7717/peerj.4731#supplemental-information.

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
