# Peer review of "Occurrence of termites (Isoptera) on living and standing dead trees in a tropical dry forest in Mexico"

_PeerJ, doi:10.7717/peerj.4731_

## Round 0.1 · original submission · Major Revisions

You manuscript "Interaction of termites (Isoptera) with living and standing dead trees in tropical fry forest in Mexico"have been read by me and three independent referees, whose reports are enclosed. All the reviewers recognize the interest of the study, but they point out that the manuscript needs careful revision in order to be acceptable for publication and I fully agree with them. The manuscript needs major rework according to reviewers’ comments before being suitable for publication in PeerJ.

Reviewer 1 ·

Basic reporting

The English language in this submission needs considerable improvement before it is suitable for publication in PeerJ. I would encourage the authors to find a qualified English Language editing service that is familiar with science writing to edit the entire manuscript. The current version has numerous - too many to point out - grammatical errors.
Listed below are some examples but the only remedy is to have someone edit the entire submission prior to returning this manuscript for consideration;
Lines 14-16 - "As ecosystem engineers, termites play a key role in ecological processes, although its functional role in the dynamics of woody material degradation in tropical dry forests has been poorly studied, particularly at canopy level." Should read more like… 'Termites play a key role as ecosystem engineers in numerous ecological processes although their role in the dynamics of wood degradation in tropical dry forests, particularly at the level of the crown canopy, has been little studied.'
Lines 22-25 - "We also found a high density of trees associated with termites in trees with diameters at breast height fewer than 15 cm. Interestingly, 72 % of variation in the density of trees associated with termites is explained by the density of standing dead trees." We found that a high density (I AM NOT CERTAIN WHAT THE AUTHORS MEAN BY HIGH DENSITY Figure 3 does not address plot density only density of living and dead trees with and without termites BUT THAT TERM SHOULD BE FOLLOWED BY A VALUE OR RANGE OF VALUES such as… 0.03-0.08 trees/m2 ) of trees with a Diameter at Breast Height (DBH) less than 15-cm in a plot was associated with the presence of termites in those trees.
Lines 49-51 - "Dead woody material also known as coarse woody debris is an important component of decomposition process since it represent as much as 42% of above-ground biomass in tropical forests" Should read more like… Coarse woody debris (CWD) is an important dead-wood component of tropical forests, and therefore the decomposition process, that can represent as much as 42% of the above-ground biomass (citations)
Lines 66-68 - "In this study we analyzed the interaction of termites with living and standing dead trees of the tropical dry forest at Chamela, Halisco, in order to understand the role of termites in the decomposition dead woody material in the canopy of a tropical dry forest of Mexico." Should read more like… In this study we recorded the presence of termites in and on living and standing dead trees in a tropical dry forest at Chamela, Halisco, in order to illuminate the role termites could play in the decomposition of dead woody material in the crown canopy of a tropical dry forest of Mexico.

Lines 72-75 - the authors must qualify this statement because standing dead wood has a lifespan - it does not remain standing forever.

The font size and character changes several times in the manuscript with the first incident occurring in Line 78. This must be corrected.

Line 111 - the authors must describe the method they used to "estimate the density" which eventually is reported as the number of trees per square meter.

Lines199-201 - could that difference be explained by periodic inundation during flooding events that would kill the termites?

Lines 236-244 - The main idea posited in these sentences should probably be placed in the Introduction to justify the authors concentration on above ground necromass. The question I had throughout reading this submission was how much of the wood in Chamela was on the ground? until I got to those lines near the end of the manuscript.

Likewise, the authors also should clearly define the term "canopy" early in the manuscript.

Experimental design

The design and statistical analysis was appropriate and the aims are within the scope of the journal.

Validity of the findings

The information provided by this work is very interesting but would have benefited from some measure of active feeding rather than the simple 'signs' of activity. It also goes without mention that some description of the termites involved - was it only one of the 30+ species found in the forest or all 30+? - would have been a major contribution.

The work is, however, important because it draws much needed attention to above ground necromass in forests and the route(s) by which that significant source of carbon is moved through the forest ecosystem.

Additional comments

This is important work that was conducted in a scientifically sound manner. Please take the time to have this work presented in an English Language journal.

Reviewer 2 ·

Basic reporting

Reporting is clear throughout. There are a few minor grammatical errors (e.g. Line 78) that should be corrected. There is sufficient coverage of the relevant literature and background literature provided. The overall structure of the article is professional, although there are numerous cases of changes in font type and size that should be changed (e.g. Lines 78-88) as these are not necessary or helpful. Raw data are shared. This is a small, correlational study and thus the questions, study design, and conclusions are self-contained and relevant to the methods and results.

Experimental design

This is original, primary research. The research questions are straight-forward and do address a research gap in termite ecology and deadwood processes that are currently largely unknown. Results are preliminary. That is, the small scope of the sampling design and the limited data collected (occurrence of termite galleries on live and dead trees) provide little information beyond whether or not termites are prevalent in standing live or dead wood in this ecosystem. That said, the results are nevertheless an interesting contribution as so little is known about termite occupancy of standing dead wood in ecosystems throughout the world. Methods are sufficiently described for replication although limited in scale and scope.

Validity of the findings

Results are narrowly novel, but nevertheless valuable to the field. This study can certainly serve as a starting point for asking broader questions about termite/standing dead wood interactions. Data are generally robust, although lacking in important details, such as type/species of trees, species of termite and preference for live/dead wood. The limited spatial scope of the sampling and lack of sampling in any other habitats other than “riparian” and “deciduous forest” suggests caution in generalizing results. Furthermore, limited description of any other important habitat impacts such as human activity levels, suggests caution in generalizing results. Conclusions are clearly stated and linked to results. I would caution against speculating too much and trying to test new hypotheses in the Discussion (Lines 212-215), but this is a minor criticism.

Additional comments

Lacking termite species, tree species, and other relevant habitat variables (e.g. land use history and role of human activity), this study is nevertheless an interesting, preliminary, contribution.

Reviewer 3 ·

Basic reporting

No comment

Experimental design

No comment

Validity of the findings

The authors need to answer: what was the interaction studied in the manuscript? It is not clear throughout the text. The presence of termite galleries on wood can not be freely interpreted as consumption. In general, the galleries on the wood usually only connect the colony to the resource source. In other words, the frequency of termite galleries in trees is different from the rate of wood consumption and also does not represent an indication of the abundance of these insects in the studied ecosystems. Therefore, the discussion is overly speculative and few supported by the results. Part of the conclusions are also not supported by the results (more details see "general comments for the author").

I suggest that the discussion and conclusion be rewritten and more focused on the results found. Speculations are welcome, but need to be made with caution because they can lead the reader to misinterpretations of the results.

Additional comments

The manuscript "Interaction of termites (Isoptera) with living and standing dead trees in a tropical dry forest in Mexico" is well written and has interesting aims.

It analysed the "interaction" (density and proposition of living and standing dead tree associate with termites) of termites with the trees in the forest canopy in two ecosystems from Mexico.

The manuscript sought to answer six questions (lines 82-88). In my opinion, all the questions were answered. However, the authors need to answer: what was the interaction studied in the manuscript? It is not clear throughout the text.

The methods and statistical analysis were appropriately used in the manuscript.

The results were well presented, using tables and figures appropriately. On the other hand, it was frustrating not to see a termite species checklist present in the trees, making it difficult to interpret the type of interaction between these insects and the vegetation. For example, because the colonies of the Kalotermitdae species live within the wood, the occurrence of these insects would undoubtedly indicate the consumption of wood in the ecosystems. On the other hand, only the presence of galleries of species of the subfamilies Nasutitermitinae and Termitinae does not make clear the type and the intensity of the existent interactions.

A considerable part of the discussion suggests a link between frequency of occurrence of termite galleries on trees with the importance of these insects in the decomposition of plant necromass. The presence of termite galleries on wood can not be freely interpreted as consumption. In general, the galleries on the wood usually only connect the colony to the resource source. In other words, the frequency of termite galleries in trees is different from the rate of wood consumption and also does not represent an indication of the abundance of these insects in the studied ecosystems. Therefore, the discussion is overly speculative and few supported by the results.

The focus of the discussion should be just the frequency of occurrence of termite galleries on trees. Any link between the gallery frequencies with the abundance or rate of wood consumption by termites needs to be done with great caution, because it may induce readers to think that galleries occurrence is directly linked to the importance of termites in the process of decomposition of plant necromass.

The same happens with the conclusions of the manuscript (see lines 266-269): "as well as the positive relationship between the density of trees associated with termites and the density of standing dead trees demonstrate that termites are important agents in wood decomposition at canopy level in this forest".

Finally, I suggest that the discussion and conclusion be rewritten and more focused on the results found. Speculations are welcome, but need to be made with caution because they can lead the reader to misinterpretations of the results.

---

## Round 0.2 · accepted · Accept

I apologize for delay in this decision. We had problems identifying reviewers, and so this revised manuscript has now been reviewed by one reviewer (the other previous referees were not available).

I have assessed your revised manuscript, and am please to let you know that it is now acceptable for publication.

# Reviewer 2 ·

Basic reporting

The authors have addressed the requested changes by the reviewers and I have no further substantive criticisms.

Experimental design

The authors have addressed the requested changes by the reviewers and I have no further substantive criticisms.

Validity of the findings

The authors have addressed the requested changes by the reviewers and I have no further substantive criticisms.

Additional comments

The authors have addressed the requested changes by the reviewers and I have no further substantive criticisms.